# Glucocorticoid Receptor β (GRβ): Beyond Its Dominant-Negative Function

**DOI:** 10.3390/ijms22073649

**Published:** 2021-03-31

**Authors:** Patricia Ramos-Ramírez, Omar Tliba

**Affiliations:** 1Department of Biomedical Sciences, College of Veterinary Medicine, Long Island University, Brookville, NY 11548, USA; Patricia.RamosRamirez@liu.edu; 2Department of Medicine, Robert Wood Johnson Medical School, Rutgers Institute for Translational Medicine and Science, New Brunswick, NJ 08901, USA

**Keywords:** glucocorticoids, GR isoforms, GRβ, metabolism, inflammation, proliferation, migration, apoptosis

## Abstract

Glucocorticoids (GCs) act via the GC receptor (GR), a receptor ubiquitously expressed in the body where it drives a broad spectrum of responses within distinct cell types and tissues, which vary in strength and specificity. The variability of GR-mediated cell responses is further extended by the existence of GR isoforms, such as GRα and GRβ, generated through alternative splicing mechanisms. While GRα is the classic receptor responsible for GC actions, GRβ has been implicated in the impairment of GRα-mediated activities. Interestingly, in contrast to the popular belief that GRβ actions are restricted to its dominant-negative effects on GRα-mediated responses, GRβ has been shown to have intrinsic activities and “directly” regulates a plethora of genes related to inflammatory process, cell communication, migration, and malignancy, each in a GRα-independent manner. Furthermore, GRβ has been associated with increased cell migration, growth, and reduced sensitivity to GC-induced apoptosis. We will summarize the current knowledge of GRβ-mediated responses, with a focus on the GRα-independent/intrinsic effects of GRβ and the associated non-canonical signaling pathways. Where appropriate, potential links to airway inflammatory diseases will be highlighted.

## 1. Introduction

Glucocorticoids (GCs) are steroid hormones produced by the adrenal cortex in response to the hypothalamic–pituitary–adrenal axis activation [1]. GCs control multiple physiological processes, including metabolic homeostasis, immune response, development, reproduction, and cognition. In addition to these physiological actions, GCs exert potent anti-inflammatory and immunosuppressive effects [2,3,4]. Because of such effects, synthetic GCs have been widely used for the treatment of inflammatory conditions, such as rheumatoid arthritis, inflammatory bowel disease, and asthma [5,6], and in the prevention of organ transplant rejection [3,7].

GCs act via the GC receptor (GR), a ubiquitously expressed receptor that drives a broad spectrum of responses within distinct cell types and tissues, which vary in strength and specificity [1,4]. The variability of GR-mediated cell responses is further extended by the existence of multiple GR isoforms generated from either alternative translation initiation or alternative splicing [1,2]. Alternative splicing near the end of the primary GR transcript (exon 9) generates two major isoforms that are different at their carboxyl-terminal sequences, GRα and GRβ [8,9]. While GRα is the most abundant GR isoform and is the classic receptor responsible for GC actions, GRβ has been implicated in the impairment of GRα-mediated activities [10,11]. 

While the GRα-dependent effects of GRβ gained a lot of attention, the GRα-independent effects of GRβ received less consideration. Indeed, in contrast to the popular belief that GRβ actions are restricted to its dominant-negative effects on GRα-mediated responses, GRβ has been shown to have intrinsic activities and “directly” regulates a plethora of genes related to inflammatory process, cell communication, migration, and malignancy, each in a GRα-independent manner [12,13,14]. Furthermore, GRβ has been shown to increase cell migration and growth and to reduce cell sensitivity to GC-induced apoptosis [15,16,17]. This report summarizes the current knowledge of GRβ-mediated responses, with a focus on GRα-independent/intrinsic effects of GRβ and the associated non-canonical signaling pathways influenced by GRβ activities. Where appropriate, potential links to airway inflammatory diseases will be highlighted and their potential impact will be discussed.

## 2. Structure and Mechanisms of Action of the Glucocorticoid Receptor

The human GR (hGR) belongs to the nuclear hormone receptor family and acts as a ligand-inducible transcription factor. hGR is a modular protein consisting of an N-terminal transactivation domain (NTD), a DNA-binding domain (DBD), a hinge region, and a C-terminal ligand-binding domain (LBD) [1,4,7]. The NTD contains a constitutively active ligand-independent activation function 1 (AF-1) domain, which allows the binding of transcriptional co-regulators. The DBD possesses two zinc finger motifs that permits the binding to specific DNA sequences in the promoters of GC target genes, the glucocorticoid-responsive elements (GREs). The LBD consists of a globular structure shaped by 12 α-helices that forms a central pocket for GC binding. Additionally, LBD contains the ligand-dependent activation function 2 (AF-2) domain, which interacts with coactivators or corepressors. The C-terminal LBD is separated from the DBD by a hinge region, which facilitates the formation of GR dimers. Moreover, the DBD/hinge region and the LBD contain nuclear localization signals (NLSs), which mediate the translocation of GR to the nucleus [1,2,18,19]. In the absence of a ligand, GR is localized in the cytoplasm forming a large inactive complex with chaperone proteins, including the heat shock proteins Hsp90 and Hsp70, p23, and the immunophilins FKBP51 and FKBP52 [2,18]. Upon binding of a ligand to the LBD region of monomeric GR, FKBP51 is exchanged by FKBP52, which exposes the NLSs and facilitates the translocation of the GR/ligand complex to the nucleus [3,18]. Once in the nucleus, the GR binds as homodimers to specific GREs and regulates the transcription of a variety of GC target genes through two main mechanisms, transactivation and transrepression. GRα-mediated transactivation activities induce gene expression through the direct binding of GRα to GRE sequences, either alone or in association with other transcription factors (TFs) [3,19]. GRα-mediated transrepression activities repress gene expression either directly where GRα binds to negative GREs (nGREs) or indirectly where GRα physically interacts with different TFs to interfere with their abilities to bind their corresponding DNA binding sites [19,20,21].

## 3. GRβ Isoform

### 3.1. Transcriptional Induction of GRβ 

The hGR is encoded by the *NR3C1* gene (nuclear receptor 3, group C, member 1) located on chromosome 5 (5q31Y32) and is composed of nine exons. Alternative splicing near the 3′ UTR of the primary transcript generates GRα and GRβ isoforms; while the C-terminal end of GRα derives from the proximal portion of exon 9, the C-terminal end of GRβ derives from an alternative splice acceptor site in the distal portion of exon 9. Since exon 9 encodes the LBD, these isoforms differ significantly in their ligand-binding abilities. Indeed, while both GR isoforms are identical through amino acid 727, which includes NTD, DBD, and a part of the LBD, the rest of the LBD differs between the two isoforms. While GRα has an additional 50 amino acids that encode helices 11 and 12, necessary to form a hydrophobic pocket critical for the binding to GCs [1,2], GRβ has only an additional 15 non-homologous amino acids. Consequently, GRβ-LBD is shorter and lacks helix 12, thereby preventing GRβ from binding to the GCs [14]. Similar to GRα, the GRβ isoform is expressed ubiquitously in most tissues but is found at lower levels than GRα. Interestingly, the relative expression levels of GRα and GRβ have been associated with GC insensitivity in various cells and tissues. Indeed, the ratio of GRα:GRβ expression has been shown to control GC cellular responsiveness in various cells and tissues, where higher ratios correlate with GC sensitivity while lower ratios correlate with GC resistance [16,22,23,24]. For instance, in some inflammatory diseases, GRβ expression is markedly upregulated, which reduces the GRα:GRβ expression ratio and promotes GC resistance. Here, GRβ promotes GC resistance through GRα-dependent mechanisms mainly by forming a non-transactivating heterodimer with GRα, thereby impairing GRα-mediated activities [16,22,23,24].

#### 3.1.1. Role of Serine/Arginine-Rich Proteins (SRps)

While there is evidence showing that pro-inflammatory mediators such as TNFα and IFNγ selectively increase GRβ expression in airway cells [25], the molecular mechanisms that promote GRβ expression are not well understood. Since GR isoforms are generated from the same pre-mRNA transcript via an alternative splicing mechanism, the factors that modulate this process, such as serine/arginine-rich proteins (SRps), have been identified as modulators of GRβ expression (Figure 1). For instance, Xu and colleagues found a predominance of SRp30c in human neutrophils that display high levels of GRβ [26]. The authors further showed that IL-8 treatment increased the expression of SRp30c and suggested that inflammatory mediators promote GRβ expression in neutrophils through the upregulation of such SRp [26]. Bombesin is a survival neuropeptide previously shown to upregulate GRβ expression in cancer cells [27]. Interestingly, in the presence of bombesin, Zhu and colleagues showed that SRp30c mediated the alternative splicing of GR pre-mRNA to generate GRβ mRNA in PC3 cells [27]. The authors further demonstrated that SRp30c siRNA through the reduction of GRβ expression attenuated bombesin’s antagonism of GC actions in PC3 cells [27]. Similarly, studies conducted in THP-1 cells, a human monocyte cell line, showed that the dehydroepiandrosterone (DHEA)-induced increase in SRp30c expression was associated with upregulation of GRβ [28,29] (Figure 1).

Interestingly, in human trabecular meshwork (TM) cells, Jain and colleagues showed that SRp20 favored the splicing of GRα, whereas SRp30c and SRp40 favored the splicing of GRβ [30]. The addition of bombesin to TM cells enhanced the expression of SRp30c and SRp40 together with GRβ while rapidly decreased the expression of SRp20. These specific differences were associated with the predicted binding sites for SRps on exon 9 of the GR gene, with more SRp20 sites on exon 9β and more SRp40 sites on exon 9α [30,31] (Figure 1). Similarly, Yan and colleagues showed that SRp40 promotes GRβ expression in HeLa cells, where SRp40 siRNA significantly increased the ratio of endogenous GRα/GRβ in HeLa cells; however, such an effect was not observed in human embryonic kidney 293T cells [36].

Altogether, these data indicate that SRps influence the alternative splicing of GR pre-mRNA and regulate the GRα/GRβ ratio in a cell-dependent manner. Moreover, other molecular factors, such as microRNAs (miRNAs), could also control the expression of GR isoforms [37,38].

#### 3.1.2. Role of miRNAs

miRNAs are small non-coding RNA molecules that have emerged as key post-transcriptional regulators of gene expression [39,40]. Interestingly, several studies have reported the involvement of miRNAs in GR expression in different pathologies [32,37,41]. For instance, Ledderose and colleagues showed that steroid-resistant T cells obtained from sepsis patients exhibited increased miR-124 and GRβ expression when compared to T cells obtained from healthy donors [33]. The effect of miR-124 on GRβ expression seen in these cells seems to be “indirect” through the downregulation of GRα expression [33] (Figure 1). In another study, Gao and colleagues reported that miR-29a regulated the expression of GRβ in epithelial cells obtained from patients with respiratory syncytial virus (RSV) infection [41]. Using in silico prediction software, Hinds’ research group reported that three miRNAs targeted the 3′UTR region of the human GRβ gene, namely, miR-33a, miR-181a/b/c/d, and miR-144. When bladder cancer cells were transfected with the pMirTarget 3′ UTR hGRβ mutant for miR-144, a dramatic reduction in GRβ expression was observed, indicating that miR-144 promotes GRβ expression [15]. In line with this, overexpression of miR-144 resulted in a significant increase in the expression of GRβ but not of GRα [15]. Moreover, to prevent miR-144 from binding to the GRβ 3′ UTR gene region, Hinds and colleagues developed a peptide nucleic acid (PNA) conjugated with a cell-penetrating peptide (CPP), termed Sweet-P [15,34]. When T24 human bladder cancer were treated with Sweet-P, the expression of the GRβ mRNA was dramatically reduced [15] (Figure 1). Together, these findings suggest that miRNAs regulated GRβ expression in a cell- and isoform-specific manner.

### 3.2. GRβ Expression in Animal Species

In addition to humans, the GRβ isoform is expressed in several animal species (Figure 2); however, the GRβ protein sequence and gene organization (e.g., the presence of a specific donor splice site in exon 9β) are only conserved among primates [42,43] (Figure 2A). Other placental mammals, such as rodents, cats, dogs and hedgehogs, contain a distinct splice acceptor site for GRβ expression [42,44]. For instance, GRβ expression in the mouse has long been controversial where two major/pioneer studies dramatically influenced studies on murine GRβ (mGRβ). Initially, mGRβ did not receive a lot of attention, as an earlier report by Otto and colleagues demonstrated the absence of GRβ in mice [45]. It was only until 2010 that mGRβ began to gain a significant consideration when Hinds and colleagues demonstrated its expression in mice [44]. The discrepancy between these two studies is mainly due to the location of the alternative splicing site involved in the generation of the mGRβ isoform. With the thought that a similar alternative splicing process occur in humans and mice, Otto and colleagues examined the genomic region around exon 9 in mice and failed to detect any mGRβ mRNA expression [45]. Strikingly, Hinds and colleagues demonstrated the mGRβ isoform arises from a distinct alternative splicing mechanism, utilizing intron 8 rather than exon 9 as in humans [44] (Figure 2B). Such splicing produces a form of GRβ containing a C-terminal end of 15 amino acids that is similar in function as in humans, specifically in terms of its dominant-negative activities on murine GRα-mediated functions. Likewise, DuBois and colleagues demonstrated that alternative splicing of GRβ occurs in the rat by intron inclusion where both isoforms were expressed at different levels in different tissues [46]. In zebrafish, GRβ (zGRβ) is generated through alternative splicing occurring at exon 8 (Figure 2C). Interestingly, when COS-7 cells were transfected with the zGRβ expression vector, zGRα-mediated transactivation activity was significantly reduced [42]. Conversely, a recent study showed no evidence for a dominant-negative activity of zGRβ on zGRα-mediated functions when a zebrafish PAC2 cell line was used [43]. However, when specific zGRβ overexpression was achieved by injecting zebrafish embryos with zGRβ mRNA, zGRβ exerts its dominant-negative activity only when GRα expression was reduced simultaneously [47].

### 3.3. Subcellular Localization of GRβ

GRβ was originally described as an orphan receptor constitutively localized in the nucleus [11,48,49]. Interestingly, several studies have indicated that GRβ localization might be cell-dependent. Li and colleagues, for instance, showed that the cell distribution of hGRβ was different in monocytes versus T cells. Whereas GRβ was found to be localized in the cytoplasm and the nucleus of monocytes, GRβ was exclusively localized in the nucleus of T cells [50]. These authors further showed that GRβ is expressed in the nucleus of CD19^+^ B cells and CD56^+^ natural killer cells at similar level than that of T cells [50]. A very comprehensive study conducted by Schaaf and colleagues showed that either YFP-tagged hGRβ or YFP-tagged zGRβ transfected into COS-1 cells were mainly localized in the nucleus in the absence of the ligand. Interestingly, the human GR isoforms displayed a more cytoplasmic localization in the absence of a ligand than their counterpart in zebrafish [42].

The effect of GCs on GRβ sub-cellular trafficking is also cell-specific. For example, dexamethasone treatment of COS-1 cells transfected with hGRβ cDNA showed that hGRβ primarily resided in the nucleus [11]. In HeLa cells, immunostaining studies revealed that two-thirds of GRβ-positive staining was observed in the cytoplasm, while the remaining staining was seen in the nucleus [51]. All GRβ translocated into the nucleus within 30 min of dexamethasone treatment [51]. The authors further showed that, in the absence of GC, GRβ was bound to Hsp90 to form a complex mainly localized in the cytoplasm; the addition of GC interfered with such binding and released GRβ, allowing it to form a heterodimer with ligand-GRα and translocated into the nucleus [51,52] (Figure 3). Conversely, Lewis and colleagues reported no effect of dexamethasone on GRβ subcellular trafficking, both in COS-1 and U-2 OS cells [12].

Importantly, several studies reported the ability of the GR antagonist, mifepristone (RU486), to bind GRβ (Figure 3). For example, confocal microscopy studies in COS-7 and U-2 OS cells showed that RU486 was not only able to bind to GRβ but also to induce its nuclear translocation independently from any interaction with GRα [12]. In contrast, others failed to show any effect of RU486 on GRβ nuclear translocation in HCT1 16 cells [13] or in mouse embryonic fibroblast (MEF) [44], suggesting a cell-specific effect of RU486 on GRβ sub-cellular trafficking.

## 4. Physiological and Pathologic Functions of GRβ

### 4.1. GRα-Independent/Intrinsic Effects of GRβ on Gene Expression

GRβ is frequently associated with GC insensitivity in a large number of inflammatory disorders, where its ability to regulate gene expression has mainly been accredited to its antagonism of GRα [2]. However, several studies showed that GRβ can directly induce and repress a large number of genes independently of GRα antagonism [12,13,53] (Figure 3). Indeed, despite GRβ lacking the helices necessary to form the ligand-binding pocket, experimental evidence has reported that GRβ might exert transcriptional activity on several genes, including with GRE-containing promoters [12,13,52,53]. Such effects of GRβ are called “intrinsic” activities. An elegant study conducted by He and colleagues demonstrated the intrinsic transcriptional activities of GRβ using animals models [54], where C57BL/6 mice were injected with adeno-associated virus (AAV) expressing hGRβ, AAV-GFP (AAV backbone), or PBS. Whole-genome microarray analyses of livers obtained from these mice revealed 2108 significantly changed genes when the AAV-hGRβ-injected mice were compared to PBS-injected mice. However, when the AAV-hGRβ-injected mice were compared to AAV-GFP-injected mice, 1916 genes specifically regulated by the expressed hGRβ were detected. Interestingly, the expression of 90% of these latter genes were uregulated These genes were involved in distinct pathologies, such as endocrine system disorders, gastrointestinal disease, immunological disease, metabolic diseases, and inflammatory response [54]. Further Ingenuity Pathway Analysis (IPA) of genes specifically regulated by hGRβ identified important pathways affected by hGRβ, most of which were associated with innate and adaptive immunity [54]. Additionally, an increase in the mRNA expression of type I and II interferons and STAT1 was also observed, suggesting a pro-inflammatory function of hGRβ in mouse livers. Further experiments also demonstrated the ability of hGRβ to bind to an intergenic GRE site located downstream of the *STAT1* gene [54].

To further assess the GRα-independent effects of hGRβ, He and colleagues used knockout mice lacking GRα specifically in the liver (GR liver knockout (GRLKO) mice) [54]. Expression of hGRβ in the liver of GRLKO mice showed that 1670 genes were exclusively regulated by hGRβ independently from GRα. These genes were involved in distinct pathologies, such as cancer, gastrointestinal diseases, infectious diseases, endocrine system disorders, and immunological disease. Interestingly, the majority of genes regulated by hGRβ in wild-type mice were dependent on the presence of mGRα (1659 genes), while many of the genes regulated by hGRβ in GRLKO mice were dependent on the loss of mGRα (1413 genes), suggesting that hGRβ gains the ability to regulate many genes when mGRα expression is lost [54]. Similarly, using a human cell line, i.e., HeLa cells stably expressing hGRβ, Kino and colleagues also showed that GRβ has intrinsic gene-specific transcriptional activities where the majority of the regulated genes were distinct from those modulated by GRα [13]. Together, these studies clearly indicate that GRβ has GRα-independent/intrinsic activities.

### 4.2. Metabolism

While the role of GRβ in physiological processes did not receive a lot of attention, recent evidence demonstrated its critical role in various physiological cell functions. Animal studies reveal that GRβ controls different metabolic functions, such as gluconeogenesis and lipid storage. For instance, the GRβ levels, but not GRα, were increased in mice and rats subjected to fasting–refeeding [44,46]. Further, in human and mouse cell lines, several reports showed that insulin upregulated GRβ protein and mRNA expression [15,17,44,55]. These data are of clinical relevance, since GCs regulate the genes associated with glucose metabolism in skeletal muscle, adipose tissue, and liver, where insulin might antagonize GRα activity possibility via GRβ upregulation [44]. This was validated by in vivo studies, where intravenous insulin injection upregulated the GRβ expression in the liver of genetically diabetic Goto-Kakizaki rats [46]. These studies further suggested that the increment in the levels of GRβ driven by insulin resistance may promote GC resistance during obesity-induced inflammation [44,46].

The involvement of GRβ in hepatic functions has been further demonstrated in obese mice. For instance, in a high-fat diet (HFD)-induced obesity model, GRβ mRNA was elevated in adipose tissue and liver, but not in skeletal muscle, indicating that GRβ might control metabolic disorders [56]. In agreement with these data, the GRβ levels were increased during adipogenesis, whereas the GRα levels were unchanged [57]. The role of GRβ in hepatic lipid accumulation was further investigated by overexpressing GRβ in the liver of mice under a standard fat diet, which caused hepatic lipid accumulation and a marked increment of serum triglycerides, thereby contributing to the pathogenesis of non-alcoholic fatty liver disease [56]. Mechanistically, when GRβ was specifically overexpressed in the mouse liver, the expression and activity levels of different signaling molecules critical in hepatic glucose metabolism were reduced. Such molecules include glucose-6-phosphatase (G6Pase), PEPCK, glycogen synthase 2 (Gys2), glycogen synthase kinase 3β (GSK3β) phosphorylation, Akt2 hepatic expression, and PPARα transcriptional activity [56]. Additional studies performed in MEF cells showed that the expression of gluconeogenic genes, such as pyruvate dehydrogenase kinase-4 (*PDK4*) and *G6Pase*, was also inhibited by GRβ [44]. Altogether, these data clearly support the influence of GRβ in the regulation of glucose metabolism through the attenuation of hepatic gluconeogenesis.

### 4.3. Inflammation

The involvement of GRβ in the modulation of inflammation has been demonstrated by different studies. For example, several reports demonstrated that GRβ was not only induced by a variety of inflammatory mediators [25,58,59] but also activated various inflammatory pathways [54,56]. For instance, hepatic-specific overexpression of GRβ increased the pro-inflammatory M1 macrophages markers, TNFα, and inducible nitric-oxide synthases, while it reduced the anti-inflammatory M2 macrophages markers, arginase 1, and FIZZ1 [56]. Notably, GRβ activates the NF-κB signaling pathway while reduces the expression of the NF-κB inhibitor, IκB [56]. Furthermore, the overexpression of hGRβ in murine hepatocytes upregulated the expression of type I and II IFNs as well as STAT1. Strikingly, such genes were also upregulated by GRβ even in GRα-deficient liver, indicating that GRβ regulates the expression of these genes in a GRα-independent manner [54]. When chromatin immunoprecipitation (ChIP) assays were performed, GRβ was shown to be recruited to intergenic GRE sites located downstream of the STAT1 gene, indicating that GRβ regulates the expression of STAT1 at the transcriptional level [54]. In line with this, GRβ has been shown to regulate the transcriptional activity of various gene promoters, such as those encoding for PPARα, NF-κB, and PTEN, in different mouse cells lines [17,56]. However, the precise mechanisms through which GRβ regulates these different signaling pathways remain to be elucidated.

Evidence from various cell types reported that GRβ differentially modulate GRα-mediated actions. For instance, a study performed in human neuroblastoma-derived BE(2)-C cells nicely demonstrated that GRβ exerts a selective dominant-negative effect on GRα-mediated transrepression activities but not on GRα-mediated transactivation activities. These findings may explain why the GC resistance seen in patients with inflammatory diseases is restricted to the immunosuppressive/anti-inflammatory effects of GC, which are driven by transrepression mechanisms, but not to the metabolic effects of GC, which are driven by transactivation mechanisms [60]. However, other studies performed in various cell lines, such as COS-1, COS-7, or HEK-293 cells, showed no dominant-negative effect of GRβ on GRα-mediated AP-1 and NF-κB repression [61,62], suggesting the GRβ effect on GRα-mediated actions is cell-specific. Strikingly, a study conducted in HeLa cells showed that GRβ, rather than acting as a dominant-negative inhibitor of GRα-mediated transrepression activities, directly repressed the transcription of *IL5* and *IL13* genes in a histone deacetylases (HDACs)-dependent manner [53]. Together, these studies suggest that GRβ modulates various inflammatory pathways in a cell-specific manner, at least in part, through its intrinsic activity.

### 4.4. Migration

GRβ has been also incriminated in the regulation of cell migration [15,63]. For instance, Yin and colleagues demonstrated that GRβ knockdown reduced the migration of glioma cells [63]. Studied in human astrocytes using an in vitro scratch assay for modeling wound healing showed that GRβ expression significantly increased following injury, while no change in GRα expression was observed [63]. Interestingly, the injury-dependent activation of astrocytes involved GRβ nuclear interaction with β-catenin, thereby enhancing the β-catenin/T-cell factor (TCF) transcriptional activities in a GSK3β-independent manner [63].

Additional studies in HeLa and U-2 OS cells showed that GRβ through its intrinsic/GRα-independent transcriptional activities modulates migration by differentially regulating the expression of genes associated with extracellular matrix (ECM)-receptor interactions, as well as regulating those involved in the actin cytoskeleton, focal adhesion, and cell communication, such as *LAMB2*, *RAP1B*, *ITGA3*, or *ITGB1* [14]. Moreover, GRβ overexpression in HTC116 cells, a human colorectal carcinoma cell line lacking endogenous GRα, upregulated the mRNA expression of S100P, which is commonly associated with the progression of metastasis [13] while downregulating the expression of other genes associated with cell migration and metastasis, such as tenascin C (*TNC*) and laminin A4 (*LAMA4*) [13]. These data suggest that the GRα-independent effects of GRβ selectively modulate the genes associated with migration, which may differentially drive progression of the metastases in malignant tumors.

In order to examine whether targeting GRβ has any anti-migratory effects, several studies sought to identify and target the factors involved in the transcriptional regulation of GRβ. For instance, Hinds’ group showed that the overexpression of miR-144 in T24 bladder cancer cells significantly increased GRβ, but not GRα [15]. Interestingly, the miR-144 and GRβ expression levels were both increased during the migration assay [15]. Strikingly, using Sweet-P, a peptide nucleic acid that specifically targets the miR-144 binding site in the 3′UTR of GRβ, Hinds’ group showed that both the expression of GRβ as well its pro-migratory effect on bladder cancer cells were significantly attenuated [15,34]. In line with this, other studies in nasopharyngeal carcinoma cells showed that depletion of miR-144 inhibited cell migration and invasion, while restoring its expression increased these tumorigenic features [64]. In contrast, reports in rectal cell carcinoma and thyroid cancer cells suggest that miR-144 may act as a tumor suppressor and interfere with cell invasion and motility by inhibiting the serine/threonine kinase ROCK1 [65] and E-cadherin suppressors, respectively [66,67]. These findings strongly suggest that miR-144, in a cell-type-specific manner, affects cell migration and invasion, at least in part, through the upregulation of GRβ expression.

Altogether, these studies clearly highlight the modulatory role of GRβ in regulating cancer cell migration, albeit in a cell-specific manner. Further studies are needed to examine whether GRβ modulate the migration of other cell types, such as airway smooth muscle, as the increased migration of these cells constitutes a key feature of asthma pathogenesis.

### 4.5. Cell Proliferation

While the anti-proliferative properties of GCs are mediated through GRα, GRβ has been linked to promoting cell growth [55,68]. Interestingly, bombesin promotes the proliferation of neuroblastoma and pancreatic cancer cells [69,70] and augments the activity of androgen receptor (AR), a critical receptor in the progression of prostate cancer [27]. Further, bombesin treatment of PC-3 prostate cancer cells interfered with the anti-proliferative effects of dexamethasone [27]. Because bombesin upregulates GRβ expression in these cells, the authors suggested that bombesin-induced GRβ upregulation interferes with GC anti-proliferative effects and promotes cell growth [27]. Moreover, Ligr and colleagues showed that GRβ expression was increased in LNCaP cells overexpressing the AR coactivator ARA70β isoform, a coactivator highly expressed in prostate cancer and has been associated with cell tumor growth [68]. Interestingly, the proliferation of these cells was significantly reduced when transfected with GRβ siRNA [68]. Similar findings were seen when other prostate cell lines, such as RC165 and DU145, were used, but to a lesser extent [68]. The authors further suggested that the pharmacological inhibition of GRβ is a promising therapeutic strategy in a subset of prostate cancer where GRβ acts as an oncogene [68,71,72].

The PI3K/Akt pathway has long been known to promote cell growth and survival signals [73]. PTEN is a phosphatase known to inhibit Akt activity and various studies reported PTEN inactivation/deletion in several cancers [15,17]. Interestingly, recent evidence showed that GRβ may directly modulate cell growth through the PTEN/PI3K/Akt-dependent signaling pathway. For instance, GRβ overexpression in 3T3-L1 cells significantly reduced PTEN promoter activity while increasing basal Akt phosphorylation along with cell growth [17]. The authors further demonstrated that the direct recruitment of GRβ to PTEN promoter negatively regulates its transcriptional activity [17]. GRβ-mediated proliferation through PTEN inhibition was also demonstrated in C2C12 myoblasts [55] where GRβ overexpression decreased the PTEN expression. Such a decrease in PTEN was associated with enhanced expression of muscle factors, such as myogenin and MyoD, which regulate the progression of myoblast to multinucleated myotubes, arguing for a potential role of GRβ in skeletal muscle proliferation and differentiation [55]. In a clinical context, these findings support the role of GRβ in muscle physiology, where it may help to counteract the side effects of long-term GC treatment on skeletal muscle atrophy [55]. It is noteworthy to mention that Zhang and colleagues showed that miR-144 promoted nasopharyngeal carcinoma cell growth by repressing PTEN, leading to the activation of the PI3K/Akt pathway [64]. Because miR-144 has been shown to increase the expression of GRβ [15] and since GRβ has been shown to reduce PTEN expression [17], it is legitimate to speculate that GRβ similarly guides cell proliferation through repression of PTEN in nasopharyngeal carcinoma cells.

The Wnt intracellular signaling pathway promotes cell growth, migration, and differentiation in several cell types and its dysregulation was shown to facilitate tumor progression. β-catenin and TCF-4 are the main signaling molecules involved in the canonical Wnt pathway [74,75]. Interestingly, several lines of evidence indicate that GRβ may directly modulate cell growth through the Wnt/β-catenin-dependent signaling pathway. For instance, Yin and colleagues showed that scratch insult-induced proliferation of human astrocytes was associated with an increased nuclear co-localization of GRβ and β-catenin [63]. Such an association did not require the upstream regulatory kinase GSK3β as Wnt I, an activator of Wnt signaling independently from GSK3β, still promotes such an association. A similar crosstalk between GRβ and the Wnt/β-catenin signaling pathway has been shown with different glioma cell lines [76]. Indeed, nude mice transplanted with GRβ-deficient U118 and Shg44 glioma cells formed smaller tumors than mice transplanted with wild-type cells, thereby demonstrating the critical role of GRβ in glioma pathogenesis [76]. Additional mechanistic studies showed that GRβ promoted glioma development by enhancing the transcriptional activity of β-catenin/TCF [63]. While the activity of the β-catenin/TCF pathway and its target gene cyclin D1 was reduced in GRβ-deficient U118 and Shg44 glioma cells, the interaction between β-catenin and TCF-4 was not affected by a lack of GRβ [76], suggesting that GRβ induced cell proliferation independently from the formation of the β-catenin/TCF-4 complex. The authors further suggested that GRβ acts as a transcription co-factor of TCF-4 in the Wnt signaling pathway to regulate glioma cell proliferation [76].

Together, these findings clearly indicate the pro-proliferative potential of GRβ in different cell types involving different signaling pathways, including the PTEN/Akt/PI3K and/or Wnt/β-catenin/TCF signaling pathways. These findings further suggest that targeting GRβ could be a promising strategy in the treatment of diseases associated with aberrant cell growth.

### 4.6. Apoptosis

GCs are frequently used in cancer treatment because of their ability to induce cell death; however, some cancer cells are resistant to the apoptotic effects of GCs [77,78]. Studies in acute lymphoblastic leukemia (ALL) cells showed that a low GRα/GRβ ratio was associated with a diminished sensitivity to GCs [79]. In line with this, Koga and colleagues showed that prednisolone induced apoptosis more efficiently in ALL cell lines only when the GRα/GRβ ratio was high [24]. Interestingly, transcriptome analysis of cancer cell lines (human osteosarcoma U-2OS and cervical carcinoma HeLa cells) transfected with GRβ expression vector showed that GRβ modulates the expression of several anti-apoptotic genes, such as B-cell lymphoma 2 (*BCL2*), and this independently of GRα [13,14].

In non-cancer cell lines, such as PBMCs derived from patients with severe asthma, the increase in GRβ expression induced by cytokines such as IL-17 and IL-23 was associated with a reduction in the pro-apoptotic effects of dexamethasone [59]. Similarly, in mouse bone marrow-derived macrophages, an LPS induced-increase in the GRβ expression was associated with diminished pro-apoptotic effects of dexamethasone [80]. In contrast, in hepatic cells, GRβ seems to prevent the anti-apoptotic effects of GC. Indeed, Liu and colleagues showed that the increased expression of GRβ in HFD mice was associated with the loss of hydrocortisone anti-apoptotic effects in hepatic cells [81]. Interestingly, the restoration of hydrocortisone anti-apoptotic effects in mice genetically deficient in ERK was associated with a decrease in the GRβ expression levels [81]. The authors further suggest that ERK activation regulates GRβ expression, which plays a pivotal role in hydrocortisone-induced apoptosis in hepatic cells [81].

Altogether, these studies suggest that the GRβ effects on apoptosis is cell-specific. However, whether such an effect of GRβ is due to its intrinsic or extrinsic activities remain to be further investigated.

## 5. Concluding Remarks

Traditional views describe GRβ as a lowly expression GR isoform, mainly located in the nucleus, not expressed in mice, and unable to bind any ligand, modulating GC actions only through a dominant-negative effect on GRα. New evidence, however, challenged these concepts and now demonstrated, albeit in cell-specific manner, that (i) the ability of GRβ to bind ligands, such as RU486; (ii) the localization of GRβ both in the cytoplasm and the nucleus; (iii) the capability of GRβ to modulate gene expression and different physiological functions through its intrinsic activities; and (iv) GRβ expression in mice. These opposing views could be due, at least partially, to the fact that earlier studies used (i) cells lacking endogenous expression of GRβ, relying mostly on ectopic expression of GRβ (U2OS, COS7), transformed cells (rat hepatoma cells or leukemia cell lines), or yeast system, making physiological and clinical interpretation/extrapolation very challenging; and (ii) cell-based transient assays, assessing the synthetic reporter gene activities, rather than the endogenous, more relevant GC-target genes. In addition, the cell-, tissue-, and species-specific expression of regulatory molecules, e.g., miRNA and SRps regulating GRβ transcriptional induction, further contributed to such controversy. Finally, the fact that an earlier report failed to demonstrate the expression of GRβ in mice, dramatically limited the studies examining the role of GRβ in vivo in murine models. This emerging concept regarding the intrinsic activities of GRβ will not only uncover new mechanisms regulating cellular and physiological functions but may also provide a promising strategy in the treatment of diseases where GRβ plays a pathogenic role.

## Figures and Tables

**Figure 1 ijms-22-03649-f001:**
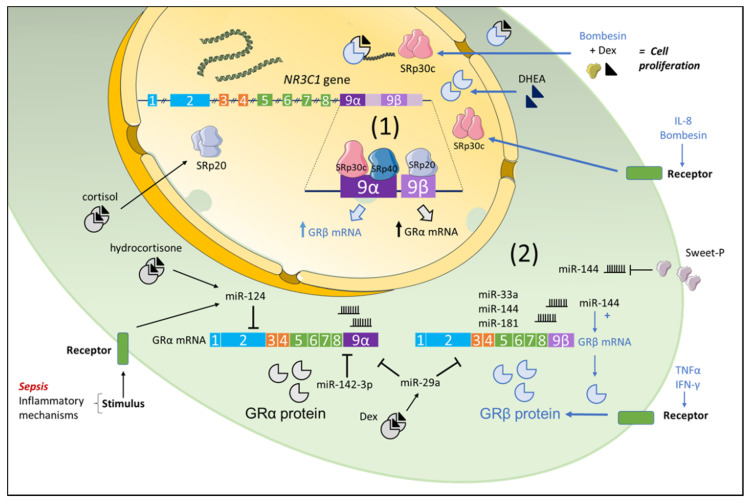
General overview of the transcriptional induction of GRβ. (**1**) Regulation of GRβ expression by serine–arginine proteins (SRps): While exon 9α is predicted to have more binding sites for SRp30c and SRp40, exon 9β possess more binding sites for SRp20. Elevated activity of SRp30c and SRp40 on exon 9α would increase exon 9β splicing and thereby the generation of GRβ mRNA. Alternatively, more binding sites for SRp20 on exon 9β would increase exon 9α and thereby the GRα mRNA splicing. Experimental evidence demonstrates that mediators, such as IL-8 in neutrophils or bombesin in PC3 cells, increase the levels of SRp30c, which enhances the alternative splicing of GR pre-mRNA to generate GRβ mRNA. Treatment with dexamethasone in the presence of bombesin resulted in increased cell number as compared to dexamethasone alone. Moreover, while dehydroepiandrosterone (DHEA) induced the upregulation of GRβ by increasing the expression of SRp30c, cortisol enhanced SRp20 to promote GRα. Therefore, changes in SRps expression result in a differential GRα/GRβ ratio. (**2**) Regulation of GRβ expression by microRNAs (miRNAs): Emerging evidence suggests that miRNAs differentially target GRα or GRβ transcripts. GRβ expression is directly regulated by targeting miRNAs that specifically binds GRβ or indirectly by miRNAs that downregulated GRα expression. While miR-33a and miR-181a/b/c/d were predicted to specifically bind the 3′UTR of human GRβ (hGRβ), where miR-144 significantly upregulated the expression of GRβ in bladder cancer cells, miR-124 and miR-142-3p specifically targeted GRα and augmented the expression of GRβ in T cells. In contrast, dexamethasone induced miR-29a, which reduced both GRα and GRβ in human and mouse adipose tissue. Inflammatory signals but also GC treatment promote the increase in miR-124 and miR-142-3p, activating a negative feedback loop that promotes GC resistance through imbalance of the GRα/GRβ ratio. Targeting miR-144 with Sweet-P significantly reduced the expression of GRβ mRNA in human bladder cancer cells. Blue arrows depict direct effects on GRβ expression, whereas black arrows depict indirect effects through the downregulation of GRα. The figure is based on References [15,25,26,27,28,29,30,31,32,33,34,35]. Parts of the figure were created using templates from Servier Medical Art, which are licensed under a Creative Commons Attribution 3.0 Unported License (http://smart.servier.com/). Last accessed on 20 January 2021.

**Figure 2 ijms-22-03649-f002:**
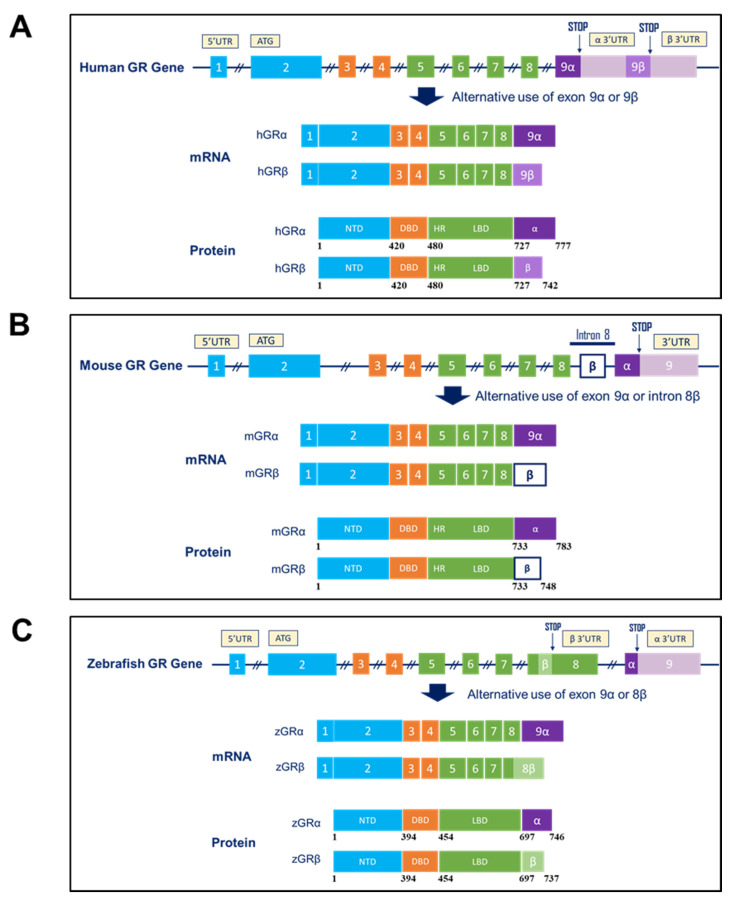
Structure of the GR gene, mRNAs, and proteins in different species. Alternative splicing of the GRβ primary transcript differs among species. (**A**) The human GR gene contains the exons 9α and 9β, which are alternatively spliced near the end of the primary transcript to generate GRα and GRβ mRNAs. GRα originates by joining the end of exon 8 to exon 9α, whereas GRβ is produced by an alternative acceptor site, where exon 8 is joined to the downstream exon 9β. The originated proteins share identical amino acids up to position 727, and thereafter, the C-terminal of GRα contains 50 additional amino acids (777 aa protein), whereas the C-terminal of GRβ contains 15 additional amino acids (742 aa protein). (**B**) The mouse GRβ (mGRβ) arises from an alternative donor site located within intron 8, rather than exon 9 as in humans. mGRα and mGRβ share identical amino acids up to position 733, and thereafter mGRβ possess 15 additional amino acids (748 aa protein). Likewise, rat GRβ originates from inclusion of intron 8. (**C**) In zebrafish, GR isoforms originate from intron retention, where zebrafish GRβ (zGRβ) arises from alternative use of a splice donor site in exon 8. The zGRβ protein contains 737 amino acids and shares the N-terminal 697 amino acids with zGRα. The figure is modified and based on References [14,42,44] and copyright permission was obtained from Oxford University Press and Copyright Clearance Center for reference [42].

**Figure 3 ijms-22-03649-f003:**
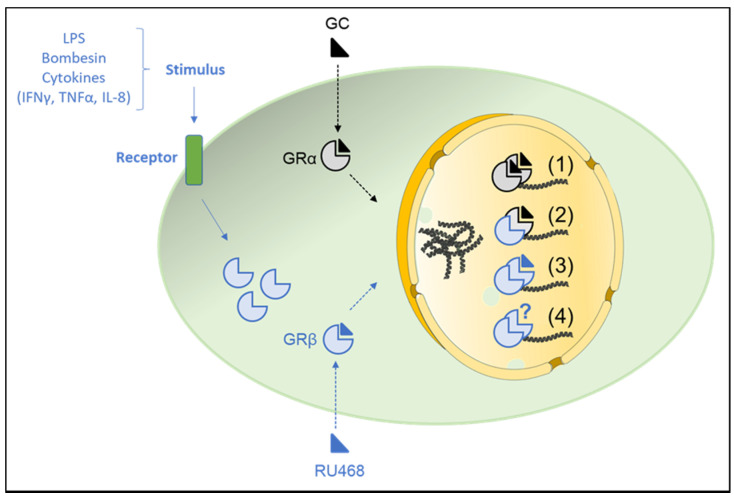
Possible mechanisms of action for GRβ. The GC function is mediated by GRα or GRβ. Different pro-inflammatory stimuli, such as TNFα, IFNγ, IL-8, or LPS, induce GRβ expression. GRβ is localized both in the cytoplasm and nucleus in a cell-type-specific manner. (**1**) Classic GC genomic effects mediated by GRα: Upon binding to GCs, GRα translocates into the nucleus where GRα homodimers bind to GREs to increase or decrease gene transcription. (**2**) GRβ dominant-negative activity on GRα: GRβ can translocate into the nucleus in a ligand-independent manner to either compete for GRE binding via their shared DBD or form inactive heterodimers with GRα. (**3**) GRβ binds RU486 to control gene expression: The synthetic GC antagonist RU486 might bind to GRβ and induce its nuclear translocation, where it modulates transcriptional activity independently of GRα. (**4**) Intrinsic activity of GRβ (unknown ligand): GRβ directly modulates the expression of a large number of genes independently of GRα. Hypothetical endogenous steroids might be involved in the nuclear translocation and the intrinsic activities of GRβ. Parts of the figure were created using templates from Servier Medical Art, which are licensed under a Creative Commons Attribution 3.0 Unported License (http://smart.servier.com/). Last accessed on 20 January 2021.

## Data Availability

Not applicable.

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
