# Peer review of "Glucocorticoid Receptor β (GRβ): Beyond Its Dominant-Negative Function"

_ijms, 2021, doi:10.3390/ijms22073649_

Round 1

Reviewer 1 Report

This article summarizes current knowledge about glucocorticoid beta receptors (GRbeta). This is a review article.

Topic:

The topic is not wide popular, but interesting. The authors rightly point out the complexity of the effect of GRbeta receptors. Their activity modification might represent a new way of therapeutic intervention into a number of important physiological and pathological conditions such as inflammation, wound healing, apoptosis, etc.; including serious diseases. Thus, the GRbeta receptors and their effects represent an interesting therapeutic challenge. Briefly, this review article could serve as a source of information on this topic and/or inspiration for further research including development of new drugs.

Formal things:

The paper is written clearly, and text is logically divided into chapters. The figures are well prepared and increase the article value.

Recommendations:

The proof reading is necessary, the text contains a part of template instructions (chapter 4.6. Apoptosis; page12/499-501).

Author Response

We would like to thank the reviewers for taking the time to carefully review our manuscript. We believe that by addressing the comments raised by the reviewers, we have significantly improved the quality and impact of our submission.

Please find enclosed our replies to the reviewers’ comments and the marked version of our revised manuscript.

Reviewer #1:

C1: The proof reading is necessary.

R1: To address the reviewer comment, the text was proofread accordingly.

C2: The text contains a part of template instructions (chapter 4.6. Apoptosis; page12/499-501).

R2: We thank the review for this valuable comment and we apologize for this oversight. The text in chapter 4.6. Apoptosis; page12/519-521 was now deleted.

Reviewer 2 Report

In this review by Ramos-Ramírez and Tliba, the authors review the most recent studies on the functions of GRbeta that are not dependent on GRalpha, and therefore ventures into perhaps less well known aspects of GRbeta function that might be important to not only understanding important diseases, in particular inflammatory disorders, but also glucocorticoid signaling in general. The authors discuss the genetic structure of the NR3C1 gene, which can be alternatively spliced to yield the two isoforms of GR, and then go over multiple mechanisms that regulate the expression of either GRalpha or GRbeta. These mechanisms include different miRNAs, and studies that have shown their specific effects on GRB expression. After covering expression in different animal species (which reveals interesting species-specific patterns of splicing), the authors then focus on the pathologic functions of GRbeta, breaking this major section into multiple subsections that include roles in metabolism, inflammation, cell migration (and possible roles in cancer cell tissue invasion), cell proliferation and apoptosis. Several important signaling pathways that are influenced by different levels of GRB expression are also reviewed under the “Cell Proliferation” section, which are important to cover. Overall the review is thorough and timely, especially considering the more recent discovery of GRbeta expression in mouse tissues, plus the differences between its expression and splicing in humans vs. other mammals. There are some issues that might be considered by the authors as follows:

1) Line 50, the authors write "GRβ has been associated with increased cell migration, growth.....", but the term "associated" should be more precisely described - were these changes shown to occur when GRB expression levels were increased? As this is the primary focus of this review, the lead-in paragraph should be as precise as possible.

2) Lines 56-81, under Section 2, Structure and Mechanisms of Action of the Glucocorticoid Receptor, the authors nicely describe both structural features of GRalpha and how it acts to either promote or repress gene activities, but they should emphasize that these mechanisms of action depend on the GRalpha isoform dimers, which would explain why GRbeta can act in a dominant negative fashion to block GRalpha activities. This becomes important when they describe the GRbeta functions independent of GRalpha in the next and subsequent sections.

3) Lines 85 through 88, the description of alternative splicing from the NR3C1 gene needs to be more clearly explained - suggest breaking this sentence with either a semi-colon or into two sentences - the first regarding the C-terminus of GRalpha from the proximal portion of exon 9, and then how this portion of alternatively spliced to yield GRbeta.

4) Line 93, the phrase "yielding to a truncated LBD lacking helices 11 and 12" needs to be reworded, as its awkward as written.

5) Line 96, the comment that "relative expression levels of GRalpha and GRbeta have been associated with GC insensitivity", this is a critical statement that should provide at least some detail of what the "relative" expression levels means, e.g., is this when GRbeta outcompetes GRalpha for GC, and this leads to insensitivity? As GRb is the focus of the review, how it might interfere with GRalpha functions is critical to establish (and will serve as a lead-in to the next section in which regulation of GRbeta expression levels are described).

6) Line 108, upon first mention of bombesin, the authors should define its use in this context, much like they did on Line 413-414.

7) In Figure 1, navigating the different cellular regions and the two major processes, transcription for either GRalpha vs. GRbeta, and then transcriptional control for each in the nucleus, is difficult especially for figuring out the protein activities, so perhaps an insert indicating the protein figures (what look like "pie portions") for each might help resolve this confusion. The large font "1" and "2" were also confusing, so could be placed in parenthesis, as done in the text, eg., (1) and (2), which will guide the reader to those descriptions.

8) Lines 239-242, the authors go into details of GRbeta subcellular localization, focusing on how expression could be modulated by GC binding and the capacity of GRbeta to form heterodimers with ligand-GRalpha, but this mechanism of modulating GRa activities via heterdimer formation with GRbeta was not adequately described in the introductory sections (see comment #2 above). Perhaps the authors could begin this section by reiterating the importance of GRa/GRb interactions, and how GRbeta can act in a dominant negative fashion to influence GRalpha activities.

9) Lines 279 - 281, did the livers from animals overexpressing hGRbeta show 1916 differentially expressed genes, or that hGRbeta directly regulated these genes? The wording implies that upon AAV-mediated expression of hGRbeta, the genes showed modulated expression, but this does not show that they were specifically regulated by hGRbeta (perhaps check reference 54).

Minor issues (please note that these are the more important issues, but that there were other minor grammatical errors, so suggest a careful review of the English and grammar):

Line 18, "... and this in a GRa-independent manner" would be better described as "each in a GRa-independent manner".

Line 36, instead of "a receptor ubiquitously expressed in the body, where it drives a...", perhaps better would be "a ubiquitously expressed receptor that drives a....".

Line 54, after "non-canonical signaling pathways" might add "influenced by GRb activities".

Line 274, the phrase "despite GRB lacks of helices....." should be edited, perhaps to "despite GRb lacks the helices necessary to form....".

Line 320, the word "no" should be "not", as in "but not in skeletal muscle....".

Lines 350-351, the authors should insert "those encoding....", after "various gene promoters, such as", since the line refers to the capacity of GRb to bind the promoter of genes that encode the listed proteins (PPARa, NF-kB, etc).

Line 382, perhaps change "the regulation of actin cytoskeleton...." to "as well as regulating those involved in actin cytoskeleton, focal adhesion.....".

Lines 499-501, the last two sentences should be removed.

Author Response

We would like to thank the reviewers for taking the time to carefully review our manuscript. We believe that by addressing the comments raised by the reviewers, we have significantly improved the quality and impact of our submission.

Please find enclosed our replies to the reviewers’ comments and the marked version of our revised manuscript.

Reviewer #2:

Major comments:

C1. Line 50, the authors write "GRβ has been associated with increased cell migration, growth.....", but the term "associated" should be more precisely described - were these changes shown to occur when GRB expression levels were increased? As this is the primary focus of this review, the lead-in paragraph should be as precise as possible.

R1. We thank the reviewer for pointing out this concern. Actually, GRb has been shown to directly stimulate and increase cell migration and growth. For clarity purposes and to address the reviewer’s comment, this sentence was rewritten and reads in lines 50-52 as follows: “Furthermore, GRb has been shown to increase cell migration and growth and to reduce cell sensitivity to GC-induced apoptosis”.

C2. Lines 56-81, under Section 2, Structure and Mechanisms of Action of the Glucocorticoid Receptor, the authors nicely describe both structural features of GRalpha and how it acts to either promote or repress gene activities, but they should emphasize that these mechanisms of action depend on the GRalpha isoform dimers, which would explain why GRbeta can act in a dominant negative fashion to block GRalpha activities. This becomes important when they describe the GRbeta functions independent of GRalpha in the next and subsequent sections.

R2. We thank the reviewer’s for this comment. To address this specific concern, we added the following sentence in lines 75-76: “Once in the nucleus, the GR binds as homodimers to the specific GREs and regulates the transcription of a variety of GC tar-get genes through two main mechanisms, transactivation and transrepression”.

C3. Lines 85 through 88, the description of alternative splicing from the NR3C1 gene needs to be more clearly explained - suggest breaking this sentence with either a semi-colon or into two sentences - the first regarding the C-terminus of GRalpha from the proximal portion of exon 9, and then how this portion of alternatively spliced to yield GRbeta.

R3. To address the reviewer’s comment and also for clarity purposes, this sentence was broke down and reads in lines 88-91 as follows: “Alternative splicing near the 3’ UTR of the primary transcript generates GRα and GRβ isoforms; While the C-terminal end of GRα derives from the proximal portion of exon 9, the C-terminal end of GRβ derives from an alternative splice acceptor site in the distal portion of exon 9”.

C4. Line 93, the phrase "yielding to a truncated LBD lacking helices 11 and 12" needs to be reworded, as its awkward as written.

R4. To address the reviewer’s comment and also for clarity purposes, few sentences were rewritten/added and read in lines 91-99 as follows: “Since exon 9 encodes the LBD, these isoforms differs significantly in their ligand binding abilities. Indeed, while both GR isoforms are identical through amino acid 727, which includes NTD, DBD and a part of the LBD, the rest of the LBD differs between the two isoforms. While GRα has an additional 50 amino acids that encode helices 11 and 12 necessary to form a hydrophobic pocket critical for the binding to GCs, GRβ has only an additional 15 non- homologous amino acids.  Consequently, GRb-LBD is shorter and lacks helix 12 preventing thereby GRb from binding to GCs.”

C5. Line 96, the comment that "relative expression levels of GRalpha and GRbeta have been associated with GC insensitivity", this is a critical statement that should provide at least some detail of what the "relative" expression levels means, e.g., is this when GRbeta outcompetes GRalpha for GC, and this leads to insensitivity? As GRb is the focus of the review, how it might interfere with GRalpha functions is critical to establish (and will serve as a lead-in to the next section in which regulation of GRbeta expression levels are described).

R5. We understand the reviewer’s concern. To address the reviewer’s comment, few sentences were rewritten/added and read in lines 100-109 as follows: “Similar to GRα, GRβ isoform is expressed ubiquitously in most tissues but is found at lower levels than GRα.  Interestingly, the relative expression levels of GRα and GRβ have been associated with GC insensitivity in various cells and tissues. Indeed, the ratio of GRα:GRβ expression has been shown to control GC cellular responsiveness in various cells and tissues where higher ratios correlate with GC sensitivity while lower ratios correlate with GC resistance. For instance, in some inflammatory diseases, GRβ expression is markedly upregulated which reduces GRα:GRβ expression ratio and promotes GC resistance. Here GRβ promotes GC resistance through GRα-dependent mechanisms mainly by forming a non-transactivating heterodimers with GRβ impairing thereby GRα-mediated activities”.

C6. Line 108, upon first mention of bombesin, the authors should define its use in this context, much like they did on Line 413-414.

R6. We thank the reviewer for pointing out this concern. The sentence in line 413-414 was now moved to lines 120-122 to define bombesin when first introduced in the text.

C7. In Figure 1, navigating the different cellular regions and the two major processes, transcription for either GRalpha vs. GRbeta, and then transcriptional control for each in the nucleus, is difficult especially for figuring out the protein activities, so perhaps an insert indicating the protein figures (what look like "pie portions") for each might help resolve this confusion. The large font "1" and "2" were also confusing, so could be placed in parenthesis, as done in the text, eg., (1) and (2), which will guide the reader to those descriptions.

R7. To address the reviewer’s comments, the suggested corrections were made accordingly and included in Fig. 1 and Fig. 3. We used different colors for the protein activities of GRa (black) versus GRb (blue).

C8. Lines 239-242, the authors go into details of GRbeta subcellular localization, focusing on how expression could be modulated by GC binding and the capacity of GRbeta to form heterodimers with ligand-GRalpha, but this mechanism of modulating GRa activities via heterdimer formation with GRbeta was not adequately described in the introductory sections (see comment #2 above). Perhaps the authors could begin this section by reiterating the importance of GRa/GRb interactions, and how GRbeta can act in a dominant negative fashion to influence GRalpha activities.

R8. We understand the reviewer’s concern. We addressed such concern in our response to comment #5.

C9. Lines 279 - 281, did the livers from animals overexpressing hGRbeta show 1916 differentially expressed genes, or that hGRbeta directly regulated these genes? The wording implies that upon AAV-mediated expression of hGRbeta, the genes showed modulated expression, but this does not show that they were specifically regulated by hGRbeta (perhaps check reference 54).

R9. We understand the reviewer’s concern. To address such concern and further clarify this specific issue, few sentences were rewritten/added and read in lines 292-300 as follows: “An elegant study conducted by He and colleagues demonstrated the intrinsic transcriptional activities of GRβ using animals models where C57BL/6 mice were injected with adeno-associated virus (AAV) expressing hGRb, AAV-GFP (AAV backbone) or PBS. Whole-genome microarray analyses of livers obtained from these mice revealed 2,108 significantly changed genes when AAV-hGRb injected mice were compared to PBS-injected mice. However, when AAV-hGRb injected mice were compared to AAV-GFP-injected mice, 1,916 genes specifically regulated by the expressed hGRb were detected. Interestingly, the expression of 90% of these latter genes were upregulated”.  

Minor comments:

C1: Line 18, "... and this in a GRa-independent manner" would be better described as "each in a GRa-independent manner".

R1. Correction was made accordingly in lines 18 and 50.

C2. Line 36, instead of "a receptor ubiquitously expressed in the body, where it drives a...", perhaps better would be "a ubiquitously expressed receptor that drives a....".

R2. Correction was made accordingly in lines 36-37.

C3. Line 54, after "non-canonical signaling pathways" might add "influenced by GRb activities".

R3. The sentence was added accordingly in lines 54-55.

C4: Line 274, the phrase "despite GRB lacks of helices....." should be edited, perhaps to "despite GRb lacks the helices necessary to form....".

R4. Correction was made accordingly in line 289.

C5. Line 320, the word "no" should be "not", as in "but not in skeletal muscle....".

R5. Correction was made accordingly in line 341.

C6. Lines 350-351, the authors should insert "those encoding....", after "various gene promoters, such as", since the line refers to the capacity of GRb to bind the promoter of genes that encode the listed proteins (PPARa, NF-kB, etc).

R6. The insertion was made accordingly in lines 371-372.

C7. Line 382, perhaps change "the regulation of actin cytoskeleton...." to "as well as regulating those involved in actin cytoskeleton, focal adhesion.....".

R7.  Correction was made accordingly in line 403.

C8. Lines 499-501, the last two sentences should be removed.

R8. Deletion was made accordingly in lines 520-522.

Reviewer 3 Report

It is rare to read a submitted manuscript written so well. The topic is interesting, despite only a few novel finding have been found in recent years. In fact, the authors describe clearly the pitfall of some old papers and some interesting development that changed the view of GRbeta function.

Minor points:

  • in my opinion, the name of miRNAs must be written with a hyphen (e.g. miR-122)
  • line 483: reasonably author wanted to write "anti-apoptotic" not "apoptotic"
  • lines 499-501: sentences must be deleted
  • line 505: reasonably author wanted to write "through" not "though" 

Author Response

We would like to thank the reviewers for taking the time to carefully review our manuscript. We believe that by addressing the comments raised by the reviewers, we have significantly improved the quality and impact of our submission.

Please find enclosed our replies to the reviewers’ comments and the marked version of our revised manuscript.

Reviewer #3:

C1: in my opinion, the name of miRNAs must be written with a hyphen (e.g. miR-122).

R1: We agree with the reviewer’s concern. Corrections were made accordingly (mainly in the figure 1) where a hyphen was added to specific miRNAs.

C2: line 483: reasonably author wanted to write "anti-apoptotic" not "apoptotic".

R2: Correction was made accordingly in line 505.

C3: lines 499-501: sentences must be deleted.

R3: We thank the review for this valuable comment and we apologize for this oversight. The text in chapter 4.6. Apoptosis; page12/520-522 was now deleted.

C4: line 505: reasonably author wanted to write "through" not "though".

R4: Correction was made accordingly in 526.